# PLEIADES: Building Temporal Kernels with Orthogonal Polynomials

**Yan Ru Pei**[*]
NVIDIA Corporation
Santa Clara, CA 95051
yanrpei@gmail.com

**Olivier Coenen**[*]
Independent Researcher
Encinitas, CA 92024
olivier@oliviercoenen.com

## Abstract

We introduce a class of neural networks named PLEIADES (PoLynomial Expansion In Adaptive Distributed Event-based Systems), which contains temporal convolution kernels generated from orthogonal polynomial basis functions. We focus on interfacing these networks with event-based data to perform online spatiotemporal classification and detection with low latency. By virtue of using structured temporal kernels and event-based data, we have the freedom to vary the sample rate of the data along with the discretization step-size of the network without additional finetuning. We experimented with three event-based benchmarks and obtained state-of-the-art results on all three by large margins with significantly smaller memory and compute costs. We achieved: 1) 99.59% accuracy with 192K parameters on the DVS128 hand gesture recognition dataset and 100% with a small additional output filter; 2) 99.58% test accuracy with 277K parameters on the AIS 2024 eye tracking challenge; and 3) 0.556 mAP with 576k parameters on the PROPHESEE 1 Megapixel Automotive Detection Dataset.

## 1  Introduction

Temporal convolutional networks (TCNs) [Lea et al., 2016] have been a staple for processing time series data from speech enhancement [Pandey and Wang, 2019] to action segmentation [Lea et al., 2017]. However, in most cases, the temporal kernel is very short (usually size of 3), making it difficult for the network to capture long-range temporal correlations. The temporal kernels are intentionally kept short, because keeping a long temporal kernel with a large number of trainable kernel values usually leads to unstable training. In addition, they require a large amount of memory to store the weights during inference. One popular solution for this has been to parameterize the temporal kernel function with a simple multilayer perceptron (MLP), which promotes stability [Romero et al., 2021] and more compressed parameters, but often significantly increases computational load.

Here, we introduce a method of parameterization of temporal kernels, named PLEIADES (PoLynomial Expansion In Adaptive Distributed Event-based Systems), that can in many cases reduce the memory and computational costs compared to explicit convolutions. The design is fairly modular, and can be used as a drop-in replacement for any 1D-like convolutional layers, allowing them to perform long temporal convolutions effectively. In fact, we augment a previously proposed (1+2)D causal spatiotemporal network [Pei et al., 2024] by replacing its temporal kernels with this new polynomial parameterization. This new network architecture serves as the backbone for a wide range of online spatiotemporal tasks ranging from action recognition to object detection.

Even though our network can be used for any spatiotemporal data (e.g. videos captured with conventional cameras), we investigate in this work the performance of our network on event-based

---

[*]Work done while at Brainchip Inc.

39th Conference on Neural Information Processing Systems (NeurIPS 2025).

data (e.g. data captured by an event camera). Event cameras are sensors that generate outputs events $\{-1, +1\}$ responding to optical changes in the scene's luminance [Gallego et al., 2020], and can generate sparse data on an incredibly short time scale, usually at 1 microsecond. Event cameras can produce very rich temporal features, capturing subtle motion patterns, and allow for flexible adjustment of the effective frames per second (FPS) in our network by varying the timebin size. This makes it suitable to test networks with long temporal kernels sampled at different step sizes. For readers interested in the performance on conventional camera data, we report preliminary results on the KITTI 2DOD task in Appendix B.3.

The code for building the structured temporal kernels, along with a pre-trained PLEIADES network for evaluation on the DVS128 dataset is available here: https://github.com/PeaBrane/Pleiades.

## 2 Related Work

### 2.1 Long Temporal Convolutions and Parameterization of Kernels

When training a neural network containing convolutions with long (temporal) kernels, it is usually not desirable to explicitly parameterize the kernel values for each time step. First of all, the input data may not be uniformly sampled, meaning that the kernels need to be continuous in nature, making explicit parameterization impossible.[2] In this case, the kernel is treated as a mapping from an event timestamp to a kernel value, where its mapping is usually achieved via a simple MLP [Qi et al., 2017, Romero et al., 2021, Poli et al., 2023]. In cases where the input features are uniformly sampled, explicit parameterization of the values for each time step becomes possible in theory. However, certain regularization procedures need to be applied [Fu et al., 2023] in practice, otherwise the training may become unstable due to the large number of trainable weights. Storing all these weights may also be unfavorable in memory-constrained environments (for edge or mobile devices).

The seminal work proposing a memory encoding using orthogonal Legendre polynomials in a recurrent state-space model is the Legendre Memory Unit (LMU) [Voelker et al., 2019], where Legendre polynomials (a special case of Jacobi polynomials) are used. The HiPPO formalism [Gu et al., 2020] then generalized this to other orthogonal functions including Chebyshev polynomials, Laguerre polynomials, and Fourier modes. Later, this sparked a cornucopia of works interfacing with deep state-space models including S4 [Gu et al., 2021a], H3 [Fu et al., 2022], and Mamba [Gu and Dao, 2023], achieving impressive results on a wide range of tasks from audio generation to language modeling. There are several common themes among these networks that PLEIADES differs from. First, these models typically only interface with 1D temporal data, and usually try to flatten high dimensional data into 1D data before processing [Gu et al., 2021a, Zhu et al., 2024], with some exceptions [Nguyen et al., 2022]. Second, instead of explicitly performing finite-window temporal convolutions, a running approximation of the effects of such convolutions is performed, essentially yielding a system with infinite impulse responses where the effective polynomial structures are distorted [Stöckel, 2021, Gu et al., 2020]. And in the more recent works, the polynomial structures are tenuously used only for initialization, but then made fully trainable. Finally, these networks mostly use an underlying depthwise structure [Howard et al., 2017] for long convolutions, which may limit the network capacity, albeit reducing the compute requirement of the network.

### 2.2 Spatiotemporal Networks

There are several classes of neural networks that can process spatiotemporal data (i.e. videos and event frames). For example, a class of networks combines components from spatial convolutional networks and recurrent networks, with the most prominent network being ConvLSTM [Shi et al., 2015]. These types of models interface well with streaming spatiotemporal data, but are oftentimes difficult to train (as with recurrent networks in general). On the other hand, we have a class of easily trainable networks that perform (separable) spatiotemporal convolutions such as the R(2+1)D and P3D networks [Tran et al., 2018, Qiu et al., 2017], but they were originally difficult to configure for online inference as they do not assume causality. However, it is easy to configure the temporal convolutional layers as causal during training, such that the network can perform efficient online inference with streaming data via the use of circular buffering [Pei et al., 2024] or incorporating spike-based [Shrestha and Orchard, 2018] or event-based [Ivanov et al., 2022] processing.

---

[2]It requires an uncountable-infinite number of "weights" to explicitly parameterize a continuous function.

## 2.3 Event-based Data and Networks

An event produced by an event camera can be succinctly represented as a tuple $E = (p, x, y, t)$, where $p$ denotes the polarity, $x$ and $y$ are the horizontal and vertical pixel coordinates, and $t$ is the timestamp of the event. A collection of events can then be expressed as $\mathcal{E} = \{E_1, E_2, ...\}$. To feed event-based data into conventional neural networks, events are often discretized into uniform timebins yielding tensors generally shaped $(2, H, W, T)$, where $H$ & $W$ are the height & width of the sensor/layer and $T$ is a time period. Different discretizations have been explored in the past. The simplest approach is to count the number of events in each timebin [Maqueda et al., 2018]. Others include replacing each event with a fixed or trainable kernel [Zhu et al., 2019, Gehrig et al., 2019] before evaluating the contribution of that kernel to a given bin. Here, we use direct binning and event-volume binning methods [Pei et al., 2024], yielding the $4d$ tensor $(2, H, W, T)$ to our network, noting that we retain the polarity unlike previous works [Zhu et al., 2019].

The most popular class of event-based networks is spiking neural networks, which generate spikes $\{0, +1\}$ with continuous timestamps at each neuron, usually described with predetermined fixed internal dynamics [Gerstner and Kistler, 2002, Gerstner et al., 2014]. These networks can be efficient during inference, as typically they only need to propagate 1-bit signals, but they are also difficult to train without specialized techniques to efficiently simulate the neural dynamics and ameliorate the spiking behaviors. The SLAYER model [Shrestha and Orchard, 2018] computes the neuron response using a kernel from the spike response model (SRM), taking only limited forms such as decaying exponent & alpha function. The kernel is used to temporally convolve the input spikes. PLEIADES generalizes the impulse-response kernel of SLAYER by making the kernel a fully trainable convolution kernel, thus fully adapted to input and network structure.

A recent line of work (in parallel to ours) is to interface event-based processing with deep state-space modeling [Schöne et al., 2024, Soydan et al., 2024], while we here still retain the orthogonal polynomial structures (see Section 2.1). Other works have proposed training using differentiable functions, surrogate gradients, to bridge the discontinuous gap of spikes to use backpropagation [Neftci et al., 2019], so that spiking networks can be trained like standard neural networks.

## 3 Temporal Convolutions with Polynomials

In this section, we discuss: 1) how the temporal kernels are generated from weighted sums of polynomial basis functions; 2) how the temporal kernels are discretized in timebins; 3) how the convolution with the input feature tensor can be optimized by changing the order of operations. In the following, we index the input channel with $c$, the output channel with $d$, the polynomial degree or basis with $n$, the spatial dimensions with $x$ and $y$, the input timestamp with $t$, the output timestamp with $t'$, and the temporal kernel timestamp with $\tau$.

### 3.1 Building temporal kernels from orthogonal polynomials

Jacobi polynomials $P_n^{(\alpha,\beta)}(\tau)$ are a class of polynomials that are orthogonal relative to a weighting function:

$$h_n^{(\alpha,\beta)} \int_{-1}^{1} P_n^{(\alpha,\beta)}(\tau) P_m^{(\alpha,\beta)}(\tau)(1-\tau)^\alpha (1+\tau)^\beta \, d\tau = \delta_{nm},$$

where $\delta_{nm}$ is the Kronecker delta, which is 1, or 0, if $n = m$, or $n \neq m$, respectively, hence establishing the orthogonality condition. $h_n^{(\alpha,\beta)}$ is a normalization constant that we ignore. A continuous function can be approximated by taking the weighted sum of these polynomials up to a given degree $N$, where the weighting coefficients $\{\gamma_0, \gamma_1, ..., \gamma_N\}$ are the trainable network parameters.

When this parameterization is used in a 1D convolutional layer typically involving multiple input and output channels, then naturally we require a set of coefficients for each pairing of input and output channels. More formally, if we index the input channels with $c$ and the output channels with $d$, then the continuous kernel connecting $c$ to $d$ can be expressed as

$$k_{cd}(\tau) = \sum_{n=0}^{N} \gamma_{cd,n} P_n^{(\alpha,\beta)}(\tau). \tag{1}$$

Training with such structured temporal kernels restricts expressivity compared to temporal kernels parametrized with a weight parameter at each timebin by not permitting discrete jumps across bins. However, there are several key advantages in using structured continuous kernels that largely overcompensate for the reduced expressivity. First, this implicit parameterization allows for resampling of the kernels during discretization, meaning that the network can interface with data sampled at different rates without additional fine-tuning (see Section 3.2). Second, having a functional basis will allow an intermediate subspace to store feature projection, which can sometimes improve memory and computational efficiency (see Section 3.3). Finally, since a Jacobi polynomial basis is associated with an underlying Sturm-Louville differential equation, this injects physical inductive biases into our network, making the training more stable and guiding it to a better optimum (see Section 5.1 for an empirical proof).

## 3.2 Discretization of the convolution kernels

In the current implementation of our network, which interfaces with time binned inputs, a discretization of the temporal kernels is needed. One method is integrating the temporal kernels over the time bins of interest.

We start by defining the antiderivative of the temporal kernels as

$$
K_{cd}(\tau) = \int_{-1}^{\tau} k_{cd}(\tau')\,d\tau' = \int_{-1}^{\tau} \sum_{n=0}^{N} \gamma_{cd,n} P_n^{(\alpha,\beta)}(\tau')\,d\tau'
$$

$$
= \sum_{n=0}^{N} \frac{\gamma_{cd,n}}{(n+1)!} P_{n+1}^{(\alpha,\beta)}(\tau) - \text{const},
$$

(2)

where the constant term does not depend on $\tau$ and can be ignored. To evaluate the integral of $k_{cd}(\tau)$ in the time bin $[\tau_0, \tau_0 + \Delta\tau]$, that we denote as the discrete $\overline{k}_{cd}[\tau_0]$, we take the difference

$$
\overline{k}_{cd}[\tau_0] = K_{cd}(\tau_0 + \Delta\tau) - K_{cd}(\tau_0)
$$

$$
= \sum_{n=0}^{N} \gamma_{cd,n} \frac{P_{n+1}^{(\alpha,\beta)}(\tau_0 + \Delta\tau) - P_{n+1}^{(\alpha,\beta)}(\tau_0)}{(n+1)!}
$$

$$
= \sum_{n=0}^{N} \gamma_{cd,n} \overline{P}_n^{(\alpha,\beta)}[\tau_0],
$$

(3)

where $\overline{P}$ is the appropriately defined discrete polynomials, to obtain the discrete form of Eq. 1. Eq. 3 can be considered a generalized matrix multiplication where the dimension $n$ (the polynomial basis dimension) is contracted, see Section 3.3. Fig. 1 provides a schematic representation of the operations to generate discretized temporal kernels for multiple channels.

Under this discretization scheme, it is very easy to resample the continuous temporal kernels (either downsampling or upsampling), to interface with data sampled at arbitrary rates, i.e. arbitrary bin sizes for event-based data. The network can be trained at a given step size $\Delta\tau$, but adapted to perform inference at a different rate (either faster or slower), without any additional tuning. The discretized polynomial basis can be regenerated using the equations above with a new $\Delta\tau$, and everything else in the network can remain unchanged.[3]

## 3.3 Optimal order of operations

Given discretized kernels, the notation is simplified by employing the Einstein notation or `einsum`, where repeating indices are summed over by convention. The equation above is rewritten as

$$
\overline{k}_{cd\tau} = \gamma_{cdn} \overline{P}_{n\tau},
$$

---

[3]This is true if the scale of the input data is invariant under resampling. For event-based data accumulated into bins, the contribution, that is, the integral over time of an event must remain independent of the discretization; the bin values have to be rescaled by a factor reciprocal to the new bin size relative to the original one. For example, if the bin size is doubled, then the bin values need to be appropriately halved.

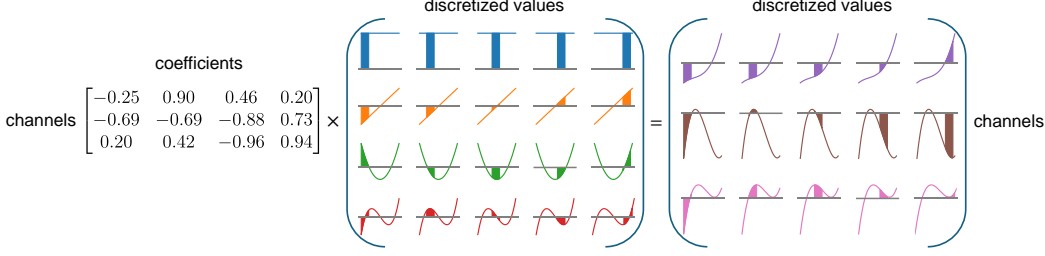

Figure 1: Generating discrete temporal kernels for multiple channels, based on trainable coefficients and fixed basis orthogonal polynomials. Here, 3 temporal kernels one per channel, is generated from 4 basis polynomials discretized over 5 timebins. The shaded areas represent the discretized polynomial values. The kernel coefficients may be organized as a $3 \times 4$ matrix, and the discretized basis polynomials may be organized as a $4 \times 5$ matrix. The matrix multiplication of the two (contraction of coefficients) then yields the final discretized kernels for the 3 channels discretized over 5 timebins as a $3 \times 5$ matrix.

where the repeating index $n$ is summed over (or contracted) on the right-hand side, corresponding to summing over the polynomial basis. See Appendix A.1 for a detailed description of the contraction rules. The temporal convolution of a kernel $\overline{k}_{ij\tau}$ with a spatiotemporal input feature tensor $u$ to obtain the output $y$ at discrete time $t'$, position $(x, y)$ and output channel $d$ is written as

$$y_{dxyt'} = \gamma_{cdn} \, M_{ntt'}(\overline{P}) \, u_{cxyt}, \tag{4}$$

where $M(\overline{P})$ is the convolution operator matrix, a sparse Toeplitz matrix generated from $\overline{P}$ (see Appendix A.3). If a depthwise convolution is performed [Howard et al., 2017], the equation simplifies to $y_{cxyt'} = \gamma_{cn} \, M_{ntt'}(\overline{P}) \, u_{cxyt}$. Here, only one channel index $c$ is needed, as the connections are parallel thus the input and output channels do not mix. The kernels are assumed to be separable into spatial and temporal convolutional kernels; thus, the temporal indices do not interact with the spatial indices $x$ and $y$. Each temporal kernel is applied separately to each spatial bin.

All einsum operations are associative and commutative,[4] so we have full freedom over the order of contractions. For example, we can first generate the temporal kernels from the orthogonal polynomials, then perform the convolutions with the input features (the typical order of operations, from left to right). However, equally valid, we can also first convolve the basis polynomials with the input features separately, then weigh and accumulate these results using the polynomial coefficients. This can be written as $\gamma_{cdn} (M_{ntt'} \, u_{cxyt}) = \gamma_{cdn} \, x_{cxynt'} = y_{dxyt'}$ in `einsum` form, where $x$ represents the intermediate projections. Note that this freedom of ordering of contractions is not possible for unstructured temporal kernels since there is no intermediate basis $n$ on which to project anything.

In practice, we select the contraction path to optimize memory or computational usage [Gray and Kourtis, 2021], depending on the training hardware and cost considerations. Memory and computational costs can be calculated for any contraction path, given the dimensions of the contraction indices (tensor shapes) (see Appendix A.2 for cost calculations). By leveraging the `opt_einsum` library, with a tailored adjustment of its cost-estimation rules (see Appendix A.3), it can automatically determine an optimal contraction path. Further refinements and applications are discussed in the related literature [Gray and Kourtis, 2021, Pei, 2025, Pei et al., 2025].

The coefficients $\gamma$ are the *trainable* parameters, and form a compressed representation of the binned temporal kernel $\overline{k}_{cd\tau}$ when the polynomial expansion is less than the number of bins $n \ll \tau$. End-to-end networks with temporal layers convolving with PLEIADES kernels can leverage standard optimization methods such as backpropagation[5]

---

[4]Einstein summation (`einsum`) restores permutation invariance to otherwise non-commutative matrix multiplications and tensor contractions by explicitly labeling contraction indices.

[5]An optimal forward contraction path automatically implies an optimal backward path, so standard autodifferentiation suffices.

# 4 Network Architecture

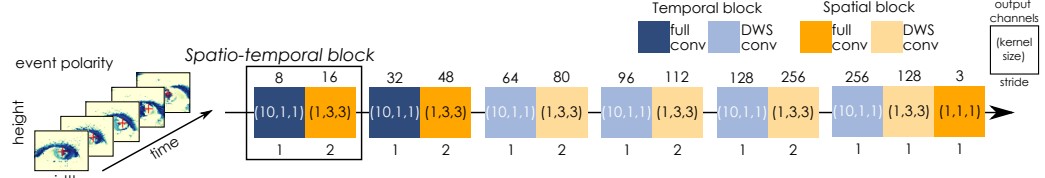

Figure 2: A representative network used for eye tracking. The backbone consists of 5 spatiotemporal blocks. Full convolutions are denoted by darker blue blocks (*full conv*), depthwise-separable convolution by lighter blocks (*DWS conv*). The detection head is inspired by CenterNet, with the modification that the $3 \times 3$ convolution is made depthwise-separable and a temporal layer is prepended to it.

The main network block is a spatiotemporal convolution block, factorized as a (1+2)D convolution. A 1D temporal convolution is performed on each spatial bin (pixel) followed by a 2D spatial convolution on each temporal frame; this is similar to a R(2+1)D convolution block [Tran et al., 2018], and the (1+2)D network for online eye tracking [Pei et al., 2024]. Each of the temporal and spatial convolutional layers may be factorized as a depthwise separable (DWS) layer [Howard et al., 2017] to further reduce computational costs. For every temporal kernel (every channel pairing, every layer, and every network variant), we use $\alpha = -0.25$ and $\beta = -0.25$ for the Jacobi polynomial basis functions, with degrees up to $4$, which were determined empirically to be suitable for all of our experiments.[6] See Appendix B.1.2 for details.

As additional notes, the choice of $(\alpha, \beta)$ does not affect the expressivity of the temporal kernels, and all choices of $(\alpha, \beta)$ yield polynomial bases spanning the same space. What it really influences here is the training dynamics, and a full mathematical justification of why $\alpha = \beta = 0.25$ yields superior training dynamics is beyond the scope of this work.[7] Nevertheless, there is a justification at an informal level. The choice of $\alpha = \beta = -0.25$ is an "intermediate" between Legendre polynomials ($\alpha = \beta = 0$) and Chebyshev polynomials ($\alpha = \beta = -0.5$), which balances the benefits of both. For instance, Legendre polynomials are more balanced, but slower to represent sharp boundary transitions; Chebyshev polynomials are better for representing edge-localized patterns, but more numerically unstable.

Key ancillary design decisions (beyond polynomial kernels) are studied more thoroughly in our previous work Pei et al. [2024]:

- **Strict causality**: All operations—including temporal convolutions—are causal, enabling low-latency online inference.

- **Hybrid normalization**: After every temporal convolution, we perform a causal Group Normalization with groups $= 4$. And after every spatial convolution, we perform a Batch Normalization. This hybrid strategy is inspired by Gordon et al. [2019].

- **Lightweight activations**: ReLU after every convolution layer and inside each DWS layer keeps implementation simple for edge devices and encourages activation sparsity.

For tasks requiring object tracking or detection (see Sections 5.2 and 5.3), we attach a temporally smoothed CenterNet detection head to the backbone (see Fig. 2), consisting of a DWS temporal layer, a $3 \times 3$ DWS spatial layer, and a final pointwise layer [Zhou et al., 2019], with ReLU activations in between. Since our backbone is already spatiotemporal in nature and capable of capturing long-range temporal correlations, we do not use any additional recurrent heads (e.g. ConvLSTMs) or temporal-based loss functions [Perot et al., 2020].

---

[6]Other choices of $\alpha$ and $\beta$ were also performant, but slightly worse than this particular choice. However, the degree choice is relatively important. Note that a degree of 10 would make the kernel essentially "free", which would actually harm performance, especially in the low-latency regime (see Section 5.1 and Appendix B.1.2).

[7]This requires extensions to theories of neural tangent kernel and dynamic mean field theory etc.

# 5 Experiments

We conduct experiments on standard computer vision tasks with event-based datasets. For all baseline experiments, we preprocess the event data into $4d$ tensors of shape $(2, H, W, T)$, with the 2 polarity channels retained. General details of data and training pipelines are given in Appendix B. With the exception of the Prophesee GEN4 experiments (Section 5.3), we run 25 trials for each experiment and report the mean and standard error (which assumes a normal distribution of noise).

## 5.1 DVS128 Hand Gesture Recognition

The DVS128 dataset (CC BY 4.0) contains recordings of 10 hand gesture classes performed by different subjects [Amir et al., 2017], recorded with a $128 \times 128$ dynamic vision sensor (DVS) camera. We use a simple backbone consisting of 5 spatiotemporal blocks. The network architecture is almost the same as that shown in Fig. 2 with the exception that the detection head is replaced by a spatial global average pooling layer followed by a simple 2-layer MLP to produce classification logits (technically a pointwise Conv1D layer during training). The output produces raw predictions at 10 ms intervals, which already by themselves are surprisingly high-quality. With additional output filtering on the network predictions, the test accuracy can be pushed to 100% (see Table 1). In addition, we compare the PLEIADES network with a counterpart that uses unstructured temporal kernels, or simply a Conv(1+2)D network [Pei et al., 2024], and find that PLEIADES performs better with a smaller number of parameters (due to polynomial compression).

Table 1: The raw 10-class test accuracy of several networks on the DVS128 dataset. Output filtering is performed only on the networks indicated by an asterisk. PLEIADES is evaluated (mean & standard error) only after all temporal layers have processed non-zero, valid frames, resulting in an inherent warm-up latency of 0.44 seconds (see Section 5.1). A 0.15 second majority filter on raw PLEIADES outputs attains 100% accuracy with minimal compute overhead, at the cost of added latency.

| Model | Accuracy | Parameters | MACs / sec |
|---|---|---|---|
| Conv(1+2)D | 99.17 | 196 K | 0.429 B |
| ANN-Rollouts [Kugele et al., 2020] | 97.16 | 500 K | 10.4 B |
| TrueNorth CNN* [Amir et al., 2017] | 96.59 | 18 M | |
| SLAYER [Shrestha and Orchard, 2018] | 93.64 | | |
| **PLEIADES** | 99.59 (0.02) | **192 K** | **0.499 B** |
| **PLEIADES + filtering*** | **100.00 (0.00)** | **192 K** | **0.499 B** |

Prior work lacks a standardized evaluation protocol: some models operate online, whereas others process entire clips before predicting. We therefore report accuracy–latency trade-off curves for each PLEIADES variant, yielding multiple Pareto frontiers. Latency is defined as the number of event frames observed since the sequence starts, multiplied by the bin size. Enforcing non-zero inputs at every temporal layer imposes an intrinsic delay of latency $= L(k - 1) \Delta\tau = 450$ ms in our baseline[8]. Zero-padding unseen frames removes this warm-up, letting the model predict after only a few bins at a slight cost in early-time accuracy. If latency is secondary, we can instead post-filter the outputs—e.g., with a causal majority or exponential filter [Amir et al., 2017]—to gain a few accuracy points, at the expense of extra delay equal to the filter length. Appendix B.1 details these latency–accuracy trade-offs.

We test two mechanisms to get the network to respond faster (Fig. 3, left). (i) **Smaller bins:** keeping the temporal kernel length fixed at $k = 10$ timebins, we reduce the bin size $\Delta\tau$ from 10 ms (baseline) to 5 ms, halving the effective temporal window. (ii) **Causal prefix masking:** during training we randomly drop an initial block of frames, forcing the model to depend on more recent inputs and shortening its functional response window.[9] Accuracy–latency traces for both strategies are plotted in the left panel; implementation details of the masking augmentation appear in Appendix B.1.

Next, we compare the previous results with keeping the kernel window constant at 100 ms and changing the bin sizes. Starting from the network trained at $\Delta\tau = 10$ ms, we resample the temporal

---

[8]Here, we used 5 layers ($L$), kernel size ($k$) of 10, and time step ($\Delta\tau$) of 10 ms.

[9]This does not alter theoretical latency but improves accuracy when fewer frames are available; it can, however, slightly hurt accuracy when the full sequence is provided.

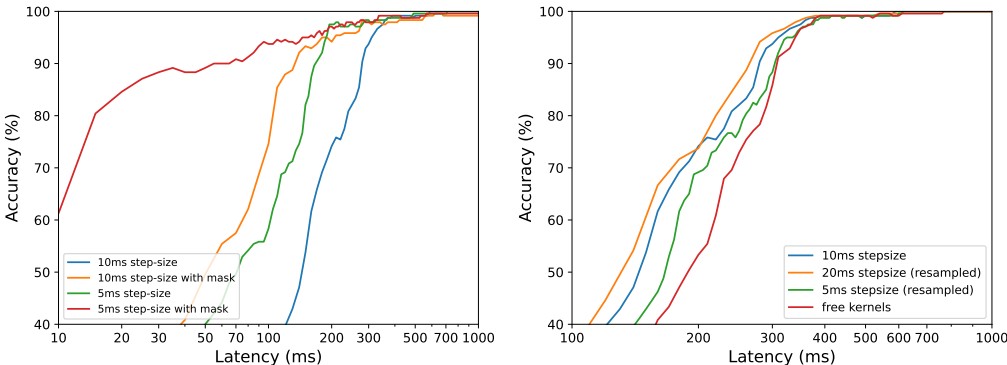

Figure 3: (Left) Accuracy vs. latency curves for different PLEIADES variants with a changing temporal window determined as a kernel size of 10 timebins but with different bin sizes on the DVS128 dataset. A masking augmentation is optionally used to randomly mask out the starting frames of dataset segments during training in order to stimulate faster responses in the network. (Right) Accuracy vs. latency curves for different PLEIADES variants with a fixed temporal window of 100 ms for each temporal layer, but having different bin sizes. The benchmark network is trained with a kernel size of 10 timebins and a 10 ms step size, and the other variants are resampled without additional fine-tuning. A network variant trained without PLEIADES structured temporal kernel is also displayed as a baseline reference (*free kernels*).

kernels without weight tuning by re-discretizing the polynomial bases (Sec. 3.2) and re-binning the events to $\Delta \tau \in \{5, 20\}$ ms, upsampling and downsampling, respectively (Fig. 3, right). The number of timesbins, $k$, is chosen so that $k \Delta \tau = 100$ ms for $k = 20$ and $k = 5$ timebins, respectively, preserving the size of the time window. The resulting accuracy–latency curves remain largely unchanged. For reference, the same plot includes a Conv(1+2)D baseline with unconstrained (*free*) non-PLEIADES kernels.

## 5.2 AIS2024 3ET+ Event-based Eye Tracking

Table 2: The 10-pixel, 5-pixel, and 3-pixel tolerances for the CVPR 2024 AIS eye tracking challenge. The performances of other models are extracted from [Wang et al., 2024].

| Model | p10 | p5 | p3 | Parameters |
|---|---|---|---|---|
| MambaPupil | 99.42 | 97.05 | 90.73 | - |
| CETM | 99.26 | 96.31 | 83.83 | 7.1M |
| Conv(1+2)D | 99.00 | 97.97 | 94.58 | 1.1M |
| ERVT | 98,21 | 94.94 | 87.26 | 150K |
| PEPNet | 97.95 | 80.67 | 49.08 | 640K |
| **PLEIADES + CenterNet** | **99.58 (0.03)** | **97.95 (0.03)** | **94.94 (0.04)** | **277K** |

The performance of our network on the 3ET+ dataset, introduced in the CVPR AIS 2024 eye-tracking challenge [Wang et al., 2024] is investigated.[10] We adopt the DVS128 hand-gesture backbone, adjusting the timebin to 5 ms. The 2-layer MLP head is replaced by a CenterNet-style detector and loss, following [Pei et al., 2024], but we predict only pupil centre points, omitting bounding-box size. See Fig. 2 for the architecture.

Table 3: Performance of PLEIADES with CenterNet detection head versus the baselines in [Perot et al., 2020] and more recent models. Frame rate (FPS) is derived from the bin size — 50 ms for the baselines and 10 ms for ours.

| Model | mAP | Parameters | MACs / sec | FPS |
|---|---|---|---|---|
| RED [Perot et al., 2020] | 0.43 | 24.1 M | | 20 |
| Gray-RetinaNet [Perot et al., 2020] | 0.43 | 32.8 M | 2060 B | 20 |
| S5-ViT-B [Zubic et al., 2024] | 0.478 | 18.2 M | | 20 |
| GET-T [Peng et al., 2023] | 0.484 | 21.9 M | | |
| **PLEIADES + CenterNet** | **0.556** | **0.576 M** | **122.5 B** | **100** |

## 5.3 Prophesee GEN4 Roadscene Object Detection

The Prophesee GEN4 Dataset is a road-scene object detection dataset collected with a megapixel event camera [Perot et al., 2020].[11] The dataset spans around 14 hours of rural/urban driving under day and night conditions. While seven classes are annotated, we follow the original protocol [Perot et al., 2020] and report mAP only for *cars* and *pedestrians*. Our detector uses a spatiotemporal hourglass backbone with a 10 ms timebin ($\Delta\tau = 10$ ms) and a CenterNet detection head (Sec. 4); no non-max suppression (NMS) is applied, as CenterNet's design and the model's temporal coherence already suppress spurious boxes. Architectural and training specifics appear in Appendix B.2. Remarkably, our model trains and infers at 100 Hz (10 ms bins) with minimal memory and compute overhead, avoiding the steep accuracy drop typically reported for high-frequency inference [Zubic et al., 2024].

## 6 Limitations

A key limitation of our design is the memory overhead incurred by the finite-window temporal filters: each layer must buffer the most recent $k$ feature maps, so the cost grows linearly with the kernel length $k$ and multiplicatively with spatial resolution. On high-resolution streams or longer kernels this cache can dominate the footprint, hindering deployment on edge devices.

Fortunately, the polynomial structure of our temporal kernels offers a principled workaround. Instead of retaining all $k$ frames, we can maintain a compact set of online basis projections — inner products between the incoming features and fixed polynomial basis functions — updated recursively at each time step [Stöckel, 2021, Gu et al., 2020]. These running coefficients play the role of hidden states in recurrent neural networks: to obtain the layer output we need only a point-wise multiplication between the coefficients and the learned kernel weights, followed by the usual spatial convolution. Conceptually, this is the forward direction of the convolution theorem — replacing convolution with a product in an appropriate transform domain.

This reformulation collapses the memory requirement from $O(kHW)$ to $O(nHW)$, where $n \ll k$ is the number of basis functions, and aligns our model with recent deep state-space architectures [Gu et al., 2021b,a]. Integrating such recurrent updates into future variants could substantially reduce memory while retaining the long temporal context that finite-window convolutions provide.

## 7 Conclusion

We presented PLEIADES, a fully-causal spatiotemporal architecture whose temporal filters are expressed as linear combinations of orthogonal polynomials. This structural choice gives the model analytically controllable receptive windows and yields *state-of-the-art* performance across all event-based vision benchmarks considered, while remaining markedly robust to time-step resampling—no additional fine-tuning was required when the bin size was halved or doubled.

---

[10]Publicly available on Kaggle; in the absence of a specific license, Kaggle's standard terms apply. The challenge organizers have confirmed that our use is permitted.

[11]Prophesee has allowed using the data in an academic context given proper citation, which we have provided.

Even in its present vanilla CNN instantiation, PLEIADES is exceptionally lean: it stores only the polynomial coefficients and small circular buffers, resulting in a memory and FLOP footprint well below contemporary counterparts. Nevertheless, two avenues could unlock further gains:

- **Activation-sparsity regularization**: Event streams are naturally sparse; injecting intermediate loss terms that penalize dense activations could cut energy consumption and latency without sacrificing accuracy.
- **Spiking conversion**: The orthogonal-polynomial filters implicitly realize high-order linear dynamics. Re-interpreting these filters as dynamical synapses would let us swap conventional ReLUs for analytically tractable spiking units whose responses exceed the expressiveness of standard leaky integrate-and-fire neurons. Such a reformulation would align PLEIADES with neuromorphic hardware, marrying convolutional-network training pipelines with the event-driven efficiency of spiking execution.

Taken together, these properties position PLEIADES as a versatile foundation for future low-latency, low-power event-based perception systems, spanning both conventional GPUs and emerging neuromorphic accelerators.

## Acknowledgments and Disclosure of Funding

We would like to acknowledge Nolan Ardolino, Kristofor Carlson, and Anup Varase (listed in alphabetical order) for discussing ideas and offering insights for this project. We would also like to thank Daniel Endraws for performing quantization studies on the PLEIADES network, and Sasskia Brüers for help with producing the figures.

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

# A    Optimal Contraction Order Memory and Compute

## A.1    The Rules of Einsum

The rules of contracting an `einsum` expression can be summarized as follows:

- At every contraction step, any two operands can be contracted together, as the `einsum` operation is associative and commutative.
- For any indices appearing in the two contracted operands but not the output and other operands, these indices can be summed away for the intermediate contraction results.

Any ordering of contractions (or a contraction path) following these rules is guaranteed to yield equivalent results.

A simple example is multiplying three matrices together, or $D_{il} = A_{ij}B_{jk}C_{kl}$. In the first stage, we can first choose to contract $A_{ij}$ and $B_{jk}$, which would yield an intermediate result of $M_{ik}$, where the index $j$ is contracted away as it does not appear in the output $D_{il}$. In the second stage, we then contract $M_{ik}$ and $C_{kl}$ to arrive at the output $D_{il}$.

We can also choose to do the contractions in any other order, and the result will remain the same. As a more extreme example, we can even perform an outer product first $M_{ijkl} = A_{ij}C_{kl}$, noting that we cannot contract away the $j$ and $k$ indices yet as they appear in $B_{jk}$ still. The contractions of $j$ and $k$ then need to be left to the second stage contraction, $D_{il} = M_{ijkl}B_{jk}$. Intuitively, we feel that this is a very suboptimal way of doing the multiplication of three matrices, and we can formalize why this is by looking at the memory and computational complexities of performing a contraction.

## A.2    Memory and Compute Requirements of a Contraction

If we assume that we are not performing any kernel fusion, and explicitly materializing and saving all intermediate tensors for backpropagation, then the extra memory and compute incurred by each contraction step is as follows:

- The memory needed to store the intermediate result is simply the size of the tensor, or equivalently the product of the sizes of its indices.
- The compute needed to evaluate the intermediate result is the product of the sizes of all indices involved in the contraction (repeated indices are counted only once).

Again, we can use the $D_{il} = A_{ij}B_{jk}C_{kl}$, where we assume that the index sizes to be[12] $i$, $j$, $k$, and $l$. Doing the first stage contraction $M_{ik} = A_{ij}B_{jk}$ will require $ik$ units of extra memory and $ijk$ units of compute, and doing the second stage contraction $D_{il} = M_{ik}C_{kl}$ will require no extra memory besides that for storing the output and $ikl$ units of compute. This gives us a total extra memory requirement of $ik$ units and a total compute requirement of $ijk + ikl$ units.

On the other hand, if we perform the outer product $M_{ijkl} = A_{ij}C_{kl}$ first, this will require $ijkl$ units of extra memory and $ijkl$ units of compute. The second stage contraction $D_{il} = M_{ijkl}C_{kl}$ will require $ijkl$ units of compute. Therefore, the total memory requirement of this contraction path is $ijkl$ units and the total compute requirement is $2ijkl$ units, both being significantly worse than the first contraction path, regardless of the sizes of the tensors.

A more subtle example (the only remaining contraction path) is contracting the operands from back to front, which we can verify requires a total memory of $jl$ units and a total compute of $jkl + ijl$ units. So this is only more memory optimal than the first contraction path if $jl < ik$, and more compute optimal if $jkl + ijl < ijk + ikl$, which may not be immediately obvious from inspection as the optimality now depends on the sizes of the tensors.

Note that it is assumed that every tensor involved in the einsum expression requires gradient from backpropagation, in the context of neural network training. This is why we identify the size of each intermediate result as "additional memory", as they need to be stored as tensors used for gradient computation. In addition, it is not difficult to see that for einsum operations, the computational costs

---

[12]From here on, we will consistently use this abuse of notation where the same letter will be used to denote both the index and the corresponding dimensional size of that index.

required for backpropagation are exactly double that of the forward computation. Therefore, we only need to consider the forward pass of the einsum expression, which is what we have been doing.

Importantly, note that this argument for memory and computational costs of gradient computations is only true under the assumption of reverse-mode automatic differentiation (backpropagation), which is what is used in almost all modern machine learning frameworks. In other words, we do not consider more general forms of automatic differentiation such as the forward-mode variant. Another important note is that in practice if the operations are memory-bound, then the computational cost estimates may not be useful for training time estimation.

### A.3 Convolution with a Parameterized Temporal Kernel

Recall in the main text that the equation for performing a full convolution with a polynomially parameterized temporal kernel is

$$y_{dxyt} = u_{cxyt}\gamma_{dnc}M_{nt't}(\overline{P}),\tag{5}$$

where the convolution operator tensor $M(\overline{P})$ based on the discretized polynomial basis functions $\overline{P}$ is given by the following Toeplitz matrix for each degree or basis $n$ (assuming that kernel size is 5 with the discretized timestamps being $\{\tau_0, \tau_1, \tau_2, \tau_3, \tau_4\}$):

$$M(\overline{P})_n =$$

$$\begin{bmatrix}
\overline{P}[\tau_0] & 0 & 0 & 0 & 0 & ... & 0 \\
\overline{P}[\tau_1] & \overline{P}[\tau_0] & 0 & 0 & 0 & ... & 0 \\
\overline{P}[\tau_2] & \overline{P}[\tau_1] & \overline{P}[\tau_0] & 0 & 0 & ... & 0 \\
\overline{P}[\tau_3] & \overline{P}[\tau_2] & \overline{P}[\tau_1] & \overline{P}[\tau_0] & 0 & ... & 0 \\
\overline{P}[\tau_4] & \overline{P}[\tau_3] & \overline{P}[\tau_2] & \overline{P}[\tau_1] & \overline{P}[\tau_0] & ... & 0 \\
 & & & ... & & & \\
0 & ... & \overline{P}[\tau_4] & \overline{P}[\tau_3] & \overline{P}[\tau_2] & \overline{P}[\tau_1] & \overline{P}[\tau_0]
\end{bmatrix},\tag{6}$$

where for a valid-type convolution we omit the first four rows of the matrix.

Note that we need to make two modifications to the memory and compute calculation rules in Section A.2 to adapt for the sparse and Toeplitz structure of the convolution matrix $M$. First is that the memory required for storing any tensor containing both $t, t'$ is guaranteed to be some form of convolution kernel, so it should only contribute a memory factor of $N_\tau$ (the kernel size) instead of $N_t N_{t'}$. Second is that any contraction of two tensors with one containing $t$ and the other containing $t, t'$ is guaranteed to be a temporal convolution, so should similarly contribute a compute factor of $N_{t'} N_\tau$ for valid-type convolutions and $N_t N_\tau$ for same-type convolutions. For our implementation, we monkey patch these modifications into the `opt_einsum` package used to provide memory and FLOP estimations of einsum expressions.

Table 4: The memory and compute requirements for each possible contraction path, where we are using a slight abuse of notation by allowing the index to represent the dimensional size of that index in the "extra memory" and "total compute" columns. The initial equation `cxyt,dnc,nt't` is always assumed. We assume here that $N_t = N_{t'}$ for simplicity (equivalent to performing same-type convolutions).

| Contraction Path | Extra Memory | Total Compute |
|---|---|---|
| `-> dnxyt,nt't -> dxyt'` | $dnxyt$ | $nxyt(dc + d\tau)$ |
| `-> ncxyt',dnc -> dxyt'` | $ncxyt$ | $nxyt(c\tau + dc)$ |
| `-> cxyt,dct't -> dxyt'` | $cxyt$ | $dnc\tau + dcxyt\tau$ |

Following the prescription given above for calculating the memory and computational requirements for performing contractions, we summarize the requirements of each contraction path for the temporal convolution in Table 4. We only consider the case for full convolutions, but the case for depthwise convolutions is analogous.

The first contraction path first contracts the input with the polynomial coefficients, then convolves the intermediate result with the basis functions. The second contraction path first convolves the input with the basis functions, then contracts the intermediate result with the polynomial coefficients.

The third contraction path first generates the temporal kernels from the polynomial coefficients and basis functions, then convolves them with the input features. In most cases, we see that the last contraction path is most memory efficient in typical cases, or when $c < dn$. However, the optimality for computational efficiency is more subtle and requires a comparison of $dn(c + \tau)$, $nc(\tau + d)$, and $dc\tau$.

# B    Details of Experiments

To convert events into frames, we choose the binning window to be 10 ms, unless otherwise specified. This time step is kept fixed throughout our network, as we do not perform any temporal resampling through the network. For the DVS128 and AIS2024 eye-tracking experiments, we perform simple direct binning along with random affine augmentations (with rotation angles up to 10 degrees, translation factors up to 0.1, and spatial scaling factors up to 1.1). For the Prophesee roadscene detection, we perform event-volume binning (analogous to bilinear interpolation), with augmentations consisting of horizontal flips at 0.5 probability and random scaling with factors from 0.9 to 1.1.

Recall that our network performs valid-type causal temporal convolutions which reduces the number of frames by (kernel size $- 1$) per temporal convolution. To avoid introducing any implicit temporal paddings to our network, we prepend extra frames (relative to the labels) to the beginning of the input segment. The total number of extra frames is then (number of temporal layers) $\times$ (kernel size $- 1$).

For all training runs, we use the AdamW optimizer with a learning rate of 0.001 and weight decay of 0.001 (with PyTorch default keywords), along with the cosine decay learning rate scheduler (updated every step) with a warmup period of around 0.01 of the total training steps. The runs are performed with automatic mixed precision (float 16) with the model `torch.compile`'d. All training jobs are done on a single NVIDIA A30 GPU.

For the total training epochs and walltimes (on a single A30 GPU):

- DVS128: 100 epochs and around 32 minutes, using a batch size of 64.
- 3ET+: 100 epochs and around 9 minutes, using a batch size of 64.
- Prophesee GEN4: 25 epochs and around 8 hours, using a batch size of 4.

The batch sizes are set such that it nearly saturates the available VRAM on the NVIDIA A30 GPU, or around 24 GB. With the exception of the Prophesee object detection experiments (Appendix B.2), we ran 25 trials for each experiment to report the mean and standard error estimators of the metrics.

## B.1    DVS Hand Gesture Recognition

Following the standard benchmarking procedure on this dataset, we only train and evaluate on the first 1.5 seconds of each trial, and filter out the "other" class where the subject performs random gestures not falling into the 10 predefined classes.

As mentioned, the network requires at least (number of temporal layers) $\times$ (kernel size $- 1$) $+ 1$ frames of inputs to guarantee that every temporal layer is processing "valid" nonzero input features. To generate output predictions with less than this number of frames, we can prepend zeros to layer inputs where needed to match the kernel size. This simulates the behavior of initializing the buffers of the temporal layers with zeros during online inference.

If the number of input frames is greater than (number of temporal layers) $\times$ (kernel size $- 1$) $+ 1$, then the network will produce more than one output prediction. If the latency budget allows, we can apply a majority filter to the classification predictions of the network, such that there is more confidence in the predictions.

To force the network to respond faster, we apply a custom random temporal masking augmentation sample-wise with $1/2$ probability. The random masking operation works by selecting a frame uniformly random from the first frame to the middle frame of the segment, then the selected frame and every frame preceding it is completely set to zero. This means that the network will be artificially biased to respond to more recent input features during inference, thereby effectively decreasing its response latency.

### B.1.1 Input Sparsity

We perform 4-bit quantization (with quantization aware training) on the gesture recognition network, and find that the network can achieve very high sparsity even without applying any regularization loss, given that it interfaces with event-based data and uses ReLU activations (which is sparsity promoting).

Table 5: Input sparsity for each layer of the gesture recognition network backbone under 4-bit quantization.

| Layer | Input Sparsity |
|---|---|
| Conv(1+2)D | 0.99 |
| Conv(1+2)D | 0.94 |
| Conv(1+2)D | 0.94 |
| Conv(1+2)D | 0.79 |
| Conv(1+2)D | 0.68 |

### B.1.2 Effect of Polynomial Degree and Jacobi Parameters

We tested the effect of polynomial degree on the performance of the network at the low-latency regime of 200 ms, where the network sees less than half the amount of data needed to fill its temporal buffers. Perhaps counterintuitively, having too high of a polynomial degree actually degraded performance (and having too low of a degree also underperforms). This highlights the importance of selecting an appropriate degree: one that is expressive enough to learn meaningful patterns from limited data, yet not so flexible that it overfits to temporal noise and fails to generalize when the full temporal context is unavailable.

Table 6: The effect of polynomial degree on the performance, for the DVS128 dataset at a latency of 200 ms, with Jacobi parameters fixed at $(\alpha, \beta) = (-0.25, -0.25)$

| Degree | Accuracy |
|---|---|
| 2 | 55.4 (0.03) |
| 4 | 73.2 (0.03) |
| 5 | 67.6 (0.04) |
| 8 | 60.2 (0.02) |
| 10 | 52.5 (0.02) |
| free | 52.3 (0.02) |

As shown in Table 7, the effect of varying the Jacobi parameters $(\alpha, \beta)$ is relatively minor compared to the impact of polynomial degree. While $(-0.25, -0.25)$ yields the highest accuracy, the differences across other parameter settings are within roughly two standard deviations, suggesting that the influence of these parameters may only be borderline significant.

Table 7: The effect of Jacobi parameters on the performance, for the DVS128 dataset at a latency of 200 ms, with degree fixed at 4.

| Jacobi Parameters $(\alpha, \beta)$ | Accuracy |
|---|---|
| $(-0.25, -0.25)$ | 73.2 (0.03) |
| $(-0.25, 0)$ | 72.9 (0.02) |
| $(-0.25, 0.25)$ | 72.8 (0.03) |
| $(0, -0.25)$ | 72.8 (0.04) |
| $(0, 0)$ | 72.7 (0.03) |
| $(0, 0.25)$ | 72.6 (0.03) |
| $(0.25, -0.25)$ | 72.9 (0.02) |
| $(0.25, 0)$ | 72.8 (0.04) |
| $(0.25, 0.25)$ | 72.6 (0.03) |

## B.2 Prophesee GEN4 Roadscene Object Detection

Following a recipe similar to the original paper, we remove bounding boxes that are less than 60 pixels in the diagonal. In addition, we perform event-volume binning which simultaneously performs spatial resizing from $(720, 1280)$ to $(160, 320)$ and temporal binning of 10 ms. For data augmentations, we perform horizontal flips at 0.5 probability and random scaling with factors from 0.9 to 1.1.

The CenterNet detection head produces feature frames where each frame is spatially shaped $(40, 80)$. Each pixel contains $7 + 2 + 2$ outputs containing 7 class logits (center point heatmaps), the bounding box height and width scales, and the bounding box center point $x_c$ and $y_c$ offsets. We perform evaluations directly on these raw predictions, without any output filtering (e.g. no non-max suppression). The network is trained on the full 7 road-scene classes of the dataset, and the mAP is evaluated on the cars and pedestrians classes, at confidence thresholds from 0.05 to 0.95 in steps of 0.05 and averaged using trapezoid integration.

See Table. 8 for details on the model architecture, which resembles an hourglass structure. Unless otherwise indicated, the temporal kernel size is assumed to be 10, causal and valid-type. The spatial kernel size is assumed to be $3 \times 3$, where the spatial stride can be inferred from the output shape of the layer. DWS denotes both the temporal and spatial layers in the Conv(1+2)D block as depthwise-separable. The BottleNeck block is similar (but not identical) to the IRB block in MobileNetV2; it is a residual block with the residual path containing three Conv2D layers with ReLU activations in between: a depthwise $3 \times 3$ Conv2D followed by a pointwise Conv2D quadrupling the channels followed by a pointwise Conv2D quartering the channels.

Before each decoder layer, the input feature is first upsampled spatially by a factor of $2 \times 2$. It is then summed with an intermediate output feature from an encoder layer that has the same spatial shape. To match the temporal shapes, the beginning frames are truncated if necessary. The remaining frames are projected with a pointwise convolutional layer (a long-range skip connection).

Table 8: The PLEIADES + CenterNet architecture used for the Prophesee dataset.

| Layer | Output Shape | Channels |
|---|---|---|
| Input | $(2, T, 160, 320)$ | |
| | | |
| **Encoder** | | |
| Conv(1+2)D | $(32, T - 9, 80, 160)$ | $2 \rightarrow 16 \rightarrow 32$ |
| BottleNeck 2D | $(32, T - 9, 80, 160)$ | $32 \rightarrow 32 \rightarrow 128 \rightarrow 32$ |
| DWS Conv(1+2)D | $(64, T - 18, 40, 80)$ | $32 \rightarrow 48 \rightarrow 64$ |
| BottleNeck 2D | $(64, T - 18, 40, 80)$ | $64 \rightarrow 64 \rightarrow 256 \rightarrow 64$ |
| DWS Conv(1+2)D | $(96, T - 27, 20, 40)$ | $64 \rightarrow 80 \rightarrow 96$ |
| DWS Conv2D | $(128, T - 27, 10, 20)$ | $96 \rightarrow 128$ |
| DWS Conv2D | $(256, T - 27, 5, 10)$ | $128 \rightarrow 256$ |
| | | |
| **Decoder** | | |
| Upsample | $(256, T - 27, 10, 20)$ | |
| DWS Conv2D | $(256, T - 27, 10, 20)$ | $256 \rightarrow 256$ |
| Upsample | $(256, T - 27, 20, 40)$ | |
| DWS Conv2D | $(256, T - 27, 20, 40)$ | $256 \rightarrow 256$ |
| Upsample | $(256, T - 27, 40, 80)$ | |
| DWS Conv2D | $(256, T - 27, 40, 80)$ | $256 \rightarrow 256$ |
| | | |
| **CenterNet Head** | | |
| DWS Conv(1+2)D | $(128, T - 27, 40, 80)$ | $256 \rightarrow 256 \rightarrow 128$ |
| pointwise Conv | $(11, T - 27, 40, 80)$ | $128 \rightarrow 11$ |

### B.3 Results on KITTI 2DOD task

See Table 9 for results of PLEIADES on the KITTI 2D object detection task compared to other standard networks optimized for conventional camera data. Our network architecture is the same used for the Prophesee GEN4 road-scene object detection task as given in Table 8.

| Model | mAP | Parameters | MACs / sec |
|---|---|---|---|
| RGBD Fusion (YOLOv2) | 0.482 | – | 349 B |
| SimCLR (ResNet50) | 0.570 | 26 M | 82 B |
| PLEIADES + CenterNet | 0.576 | 0.57 M | 18 B |

Table 9: Comparison of models in terms of mAP, number of parameters, and MACs per second. Note that the MACs/sec measure assumes an FPS of 10, which is used in the KITTI recordings.

