# OpenReview forum: "PLEIADES: Building Temporal Kernels with Orthogonal Polynomials"
_NeurIPS.cc/2025/Conference — NeurIPS 2025 poster_

### Official Review · Reviewer_iy2F · 2025-07-03

**Clarity:** 3
**Significance:** 3
**Originality:** 3
**Rating:** 4
**Confidence:** 3

**Summary:**

This paper introduces PLEIADES, a neural network architecture for online spatiotemporal processing, particularly with event-based data, by constructing temporal convolution kernels. Instead of using explicitly parameterized, discrete kernels, like temporal convolutions, PLEIADES generates them from a linear combination of orthogonal polynomial basis functions, i.e., Jacobi polynomials. Benefiting from this design, it is efficient, flexible, and high-performing.

**Questions:**

1. What is the key difference between the proposed method and previous work like Mamba, and what is the reason for the performance improvement over them?
2. How many trials do you conduct on each benchmark, for your method and other methods?
3. Could you provide experimental results on general object tracking benchmarks, such as LaSOT?

**Ethical Concerns:**

["NO or VERY MINOR ethics concerns only"]

**Final Justification:**

I appreciate the response from the authors. Since my questions are addressed, I hold the original score.

**Limitations:**

Yes

**Paper Formatting Concerns:**

No paper formatting concerns

**Quality:**

3

**Strengths And Weaknesses:**

Strengths:
1. The motivation of the paper is reasonable, and the proposed method is promising for its ability to tackle different sampling rates without fine-tuning.
2. The paper is clearly written and easy to follow.
3. The paper provides a powerful new tool and backbone for the community by setting a new state of the art on three major and diverse tasks.

Weaknesses:
1. The proposed method stems from established and well-known concepts (orthogonal polynomials, (1+2)D convolutions, event-based processing), and a short discussion could be included to help distinguish the novelty of this work.
2. The number of trials should be given for each experiment.
3. Apart from using temporal kernels, other methods, such as Mamba, also present superior performance for sequential modeling. Discussion and experiment should be incorporated for comparison.

---

> ### Author Rebuttal · Authors · 2025-07-24
>
> We thank reviewer iy2F for the positive feedback. We'd like to take the chance here to address the comments / questions.
>
> The reviewer is indeed correct that the network is based on the (1+2)D conv pattern. In the introduction, we mentioned that "we augment a previously proposed (1+2)D causal ... network by replacing its temporal kernels with this new polynomial parameterization", to highlight the key novelty here being our **polynomial parameterization of the temporal kernels**. However, we do realize this can be moved earlier in the introduction, so the novelty is immediately clear.
>
> As noted in Section 5 and in item 7 of the reproducibility checklist, we conducted **25 trials** for each experiment and reported the mean and standard error. In the result tables, the standard deviation is shown in brackets for our PLEIADES network. For the other methods, we report results as published in their respective papers, as some of these models are not open-source and were therefore not available for re-evaluation.
>
> For **comparison against SSMs**, we compared against MambaPupil for eye-tracking and S5-ViT-B for object detection, in addition to theoretical discussions in Section 2.1 and 6.
>
> In Section 2.1, we highlighted several key architectural differences of PLEIADES over existing SSMs: 1) our network is spatio-temporal, 2) we preserve the polynomial basis, 3) we have a more flexible connectivity structure. And in Section 6, we discuss how SSM designs can be incorporated into PLEIADES.
>
> The focus of our work was on event-based tasks, so we did not include conventional camera datasets in the paper. That being said, we are currently actively exploring the application of PLEIADES to RGB camera tasks such as object detection, segmentation, and planning. Initial results suggest a 5x reduction of parameters while achieving similar results compared to the smaller models in the YOLO series when benchmarked on the KITTI 2DMOT. And LaSOT is certainly a dataset that we could benchmark on in the future (along with other tracking benchmarks such as VOT, TrackingNet, GOT-10k)

---

> ### Author Response · Authors · 2025-08-05
>
> For your convenience, we would like to note that results on RGB data (KITTI 2DOD) [1] are included in our general comment above. We have re-attached them here as well.
>
> |        Model        |  mAP  | Parameters |  MACs / sec |
> |:-------------------:|:-----:|:----------:|:-----------:|
> | RGBD Fusion (YOLOv2) [2] | 0.482 |       |   349 B      |
> | Sim CLR (ResNet50) [3] | 0.570 |   26 M     |   82 B      |
> | **PLEIADES + CenterNet** | **0.576** | **0.57 M**     |   **18 B**      |
>
> [1] Geiger, A., Lenz, P., Stiller, C., & Urtasun, R. (2013). Vision meets Robotics: The KITTI Dataset. International Journal of Robotics Research, 32(11), 1231–1237. https://doi.org/10.1177/0278364913491297; https://www.cvlibs.net/datasets/kitti/
>
> [2] T. Ophoff, K. Van Beeck, and T. Goedemé, “Exploring RGB+Depth fusion for real-time object detection,” Sensors, vol. 19, no. 4, p. 866, Feb. 2019, doi: 10.3390/s19040866.
>
> [3] SimCLR: https://www.lightly.ai/datasets; Kitti 2d Object Detection Factsheet from Lightly: https://tinyurl.com/ycyphxt9

---

### Official Review · Reviewer_cGrC · 2025-07-03

**Clarity:** 2
**Significance:** 3
**Originality:** 2
**Rating:** 4
**Confidence:** 4

**Summary:**

The paper presents PLEIADES, a causal spatio-temporal network whose temporal filters are written as short Jacobi-polynomial expansions. This replaces a long 1-D convolution with just a few learned coefficients, giving wide temporal context while keeping the parameter count modest. Because the filters are continuous in time, they can be re-sampled, so one trained model can run at several frame rates without extra tuning. Inserted into a straightforward (1 + 2)D backbone, the approach scores 99.6 % on DVS128 gestures, 99.6 % on the AIS eye-tracking task, and 0.556 mAP on the Prophesee automotive data. The design is aimed at real-time, resource-constrained event-vision systems.

**Questions:**

Q1. Empirical Validation of Physical Inductive Bias
The manuscript asserts that Jacobi polynomials inject a “physical inductive bias” via their Sturm–Liouville connection, leading to “more stable training” and “better optima,” yet provides no training‐dynamics data in Section 5.1.
o	Request: Please include training‐loss curves, gradient‐norm trajectories, and convergence‐rate comparisons between PLEIADES and a free‐kernel (Conv(1+2)D) baseline.
Q2. Jacobi Parameter and Degree Sensitivity
The choice α=β=−0.25 and polynomial degree N=4 appears empirical.
o	Request: Provide a sensitivity analysis over α,β∈{−0.5,0,0.5} and degrees N∈{2,4,6,8}, with accuracy/stability results, and give guidelines for choosing these hyperparameters in new domains.
Q3. Generalization Beyond Event Vision
All experiments are on event‐camera benchmarks, leaving open whether PLEIADES applies to other temporal tasks.
o	Request: Test PLEIADES on at least one non‐event task (e.g. video‐frame classification or audio‐sequence modeling) or on longer time windows, and report performance to assess broader applicability.
Q4. Ablation of Core Components
It is unclear which design elements drive the reported gains.
o	Request: Provide ablation studies isolating (a) polynomial‐kernel parameterization, (b) (1+2)D factorization, (c) hybrid normalization, and (d) strict causality, showing each component’s effect on accuracy and latency.
Evaluation Criteria:
Meeting Q1 and Q2 with clear empirical data would significantly boost my confidence. Demonstrating cross‐domain performance (Q3) would broaden the paper’s impact. Comprehensive ablations (Q4) would clarify the key innovations and could raise clarity/originality scores.

**Ethical Concerns:**

["NO or VERY MINOR ethics concerns only"]

**Final Justification:**

Based on the additional information provided by the reviewers, my concerns have been eased and I have raised my score.

**Limitations:**

yes

**Quality:**

2

**Strengths And Weaknesses:**

Strengths
Quality:
The paper demonstrates good technical quality with rigorous mathematical foundations and comprehensive experimental validation. The use of Jacobi polynomials for temporal kernel parameterization is mathematically sound, leveraging well-established orthogonality properties and connections to Sturm-Liouville differential equations. The experimental methodology is thorough, with 25 trials per experiment, proper statistical reporting with standard errors, and testing across diverse event-based benchmarks. The resampling experiments provide particularly compelling evidence of the method's theoretical advantages, showing stable performance across different temporal discretizations without retraining.
Clarity:
The paper is generally well-written with mathematical exposition. Figure 1 effectively illustrates the core concept of generating discrete temporal kernels from polynomial coefficients. The progression from theoretical foundations (Section 3) to practical implementation (Section 4) and experimental validation (Section 5) follows a logical structure. The Einstein notation is used appropriately to simplify complex tensor operations, and the contraction optimization discussion provides valuable implementation insights.
Significance:
The work addresses an important problem in event-based computing where traditional approaches struggle with temporal modeling efficiency. The achieved performance improvements are substantial - PLEIADES reduces parameters by 62-96% compared to competing methods while maintaining or exceeding accuracy. The 100 FPS inference capability on complex detection tasks represents a significant practical advancement. The resampling capability without retraining has broad implications for deploying models across different sensor configurations and hardware platforms.
Originality:
The application of orthogonal polynomials to neural network temporal kernels is novel and theoretically motivated. While orthogonal polynomials have been used in state-space models (LMU, HiPPO, S4), their application to explicit finite-window temporal convolutions with resampling capabilities is original. The connection between polynomial structure and computational optimization through flexible contraction ordering is a creative contribution that bridges mathematical theory with practical implementation concerns.
Weaknesses
Quality:
The parameter selection for Jacobi polynomials (α=β=−0.25, degree up to 4) appears purely empirical without theoretical justification or sensitivity analysis. The paper lacks ablation studies examining the contribution of different architectural components, making it difficult to assess which design choices are critical. The comparison baselines, while reasonable, may not represent the absolute state-of-the-art in all cases, and some methods (like TrueNorth CNN from 2017) are becoming dated.
Clarity:
The paper suffers from several presentation issues that impede understanding, particularly for readers without deep expertise in numerical analysis.
	Concept-to-notation gap: The important orthogonality definition for Jacobi polynomials (lines 117-118) lacks equation numbering. The paper transitions too abruptly from the intuitive concept of "polynomial kernels" to dense Einstein-summation formulas without providing concrete examples. For instance, after introducing Jacobi polynomials in Section 3.1, the paper immediately jumps to the general form in equation (1)  without showing what a simple degree-2 kernel would look like in explicit vector form. This leaves readers struggling to map abstract symbols to concrete operations.
	Physical inductive bias inadequately supported: Section 3.1 claims that Jacobi polynomials provide "physical inductive biases" due to their connection with Sturm-Liouville differential equations, making training "more stable" and guiding networks to "better optima." However, this crucial claim is merely asserted without empirical evidence. The paper lacks training dynamics comparisons, gradient norm trajectories, or convergence analysis that would substantiate these theoretical advantages over free kernels.
	Notation inconsistencies: The manuscript inconsistently alternates between continuous kernels and discrete implementations without clearly establishing the discretization relationship. Additionally, Figure 1 labels coefficients as γ_n  while Section 4 references different variables, creating unnecessary confusion.
	Einstein notation overuse: While Einstein summation is powerful, its repeated use for both channel mixing and polynomial basis summation obscures the actual data flow and operation order. The same operations presented once in plain matrix notation or pseudocode would significantly improve readability, especially for equation (4) where multiple indices interact simultaneously.
	Incomplete mathematical exposition: Critical implementation details like numerical integration of orthogonal polynomials, gradient computation through the polynomial parameterization, and numerical stability considerations are either missing or relegated to appendices. This makes it difficult for readers to understand or reproduce the method.
These clarity issues collectively create barriers to understanding and adoption, particularly problematic given the paper's goal of providing a practical alternative to existing temporal modeling approaches.
Significance:
The evaluation is limited to event-based vision tasks, raising questions about generalizability to other temporal modeling domains. The memory limitation discussion reveals a fundamental scalability issue (linear growth with kernel length and multiplicative growth with spatial resolution) that could limit practical deployment. While a theoretical solution is proposed using recursive updates, this remains unimplemented and unvalidated. The method's advantages may diminish for very long temporal sequences where the polynomial approximation becomes less effective.
Originality:
While the specific application is novel, the work builds heavily on existing foundations (orthogonal polynomials, (1+2)D convolutions, CenterNet heads). The connection to state-space models is acknowledged but not deeply explored,  the paper could better position itself relative to this rich body of work. The claimed novelty of "structured temporal kernels" may be overstated given prior work on parameterized temporal filters in various forms.

---

> ### Author Rebuttal · Authors · 2025-07-25
>
> We thank the reviewer for their comprehensive and thoughtful review. We aim to address the comments and questions as thoroughly as possible within the scope of the rebuttal.
>
> - The choice of α = β = –0.25 is an "intermediate" between Legendre polynomials (α = β = –0) and Chebyshev polynomials (α = β = –0.5), which **balances the benefits of both**. For instance, Legendre polynomials are more balanced, but slower to represent sharp boundary transitions; Chebyshev polynomials are better for representing edge-localized patterns, but more numerically unstable. A full formal proof of why this choice is optimal is currently beyond the scope of this work, and would involve complex learning dynamics theory.
>
> - In Appendix B.1.2 we performed sensitivity analyses of the effects of polynomial degree, alpha, and beta. The main conclusion was that the choice of a degree being 4 was an important one, whereas the choice of (α, β) appeared less significant. Therefore, we believe degree 4 polynomials to be at the sweet spot being having expressivity while preventing overfitting (bias-variance tradeoff)
>
> - Besides TruthNorth, all the other networks are fairly modern. For instance, all the networks for eye-tracking were from the 2024 CVPR AIS challenge. And S5-ViT-B is a modern network from 2024 combining state-space modeling and transformers. We decided to include TruthNorth because it is a seminal network for event-based vision.
>
> - Due to space constraints, we focused on the aspects of Jacobi polynomials most relevant to our method, rather than providing a full theoretical background. As the reviewer noted, our approach integrates concepts from functional analysis, numerical approximation, and tensor operations. We aimed to present these concisely while maintaining coherence, with Figure 1 serving as a visual summary of the core ideas.
>
> - For comparison against the performance of free kernels, we showed in the right subplot of Fig. 3 that **having polynomial temporal kernels was better at almost all latency settings**. We acknowledge that we did not dig too deeply at the level of training dynamics, and thank the reviewer for pointing this out as a gap.
>
> - In Section 3.2, we clearly detailed the discretization of the continuous temporal kernels, and from there on, the kernels are referenced in their discretized form, denoted by the overline above k.
>
> - α, β, and γ actually refer to different parameters. (α, β) are parameters defining the Jacobi polynomial basis (via the weight function).  γ refers to the polynomial coefficients. This is detailed in Sections 3.1 and 3.2
>
> - We believe Einstein summation notation here actually makes the channel connectivity more apparent, and allows for cleaner notations. It is also crucial to the discussion of optimal contraction patterns as detailed in Appendix A. Additionally, it is also more directly translatable to np.einsum and torch.einsum, and has become increasingly prevalent (e.g. Mamba2). That being said, the introduction of the notation in Section 3.3 may have been too abrupt, and the paper would benefit from a longer exposition of this notation.
>
> - The beauty of PLEIADES is that we **do not require manual derivation of the backward pass** (gradient computation), say, compared to the SLAYER model [Shrestha 2018]. Everything is differentiable, so autograd (tf, torch, jax) would be able to handle everything for us, as evident in the attached source code. This is stated at the end of Section 3 as "PLEIADES kernels can leverage standard optimization methods such as backpropagation". In addition, the optimal contraction order in the forward pass also implies the optimality of the backward pass, as mentioned in footnote 4 and Appendix A.
>
> - While a formal analysis of gradient stability would be valuable, we note that standard practices—such as normalization layers and adaptive optimizers like AdamW—can obfuscate the implicit effects of our polynomial parameterization on training dynamics. Nonetheless, we agree this is an interesting direction for deeper theoretical investigation.
>
> - As the reviewer noted, memory cost scales linearly with temporal buffer size and is further multiplied by spatial dimensions—an inherent challenge for (1+2)D ConvNets. However, our network’s causal structure limits this cost to the buffer (not the full sequence), and the resampling feature allows for shorter temporal kernels via larger step sizes. While we acknowledge this bottleneck as a consideration for edge deployment, PLEIADES remains a strong candidate given its parameter and compute efficiency—often **1–2 orders of magnitude lower** than comparable models.
>
> - Even though PLEIADES builds on top of previous works, we do not think it is incremental. The usage of trainable polynomials in an (1+2)D network setting, to our knowledge, is completely novel and offers major performance and efficiency boosts over prior arts.
>
> To address the questions directly:
>
> - Q1 While we are unable to provide a comprehensive set of training dynamics analyses within the short rebuttal window, we kindly ask the reviewer to consider whether Figure 3 provides sufficient evidence that polynomial parameterization plays a meaningful role for PLEIADES.
> - Q2 We hope that Appendix B.1.2 addresses this concern. Based on our experiments across three diverse tasks, degree 4 consistently provided strong performance, suggesting it is a robust choice in practice. While we acknowledge that a more exhaustive sweep could be informative, our findings indicate that the exact choice of (α,β) has relatively limited impact compared to the degree. We believe these observations offer a reasonable justification for our selected configuration.
> - Q3 We appreciate the reviewer’s suggestion to explore conventional camera tasks. While we agree that extending PLEIADES to RGB-based domains such as object detection and segmentation is a promising direction—and one we are actively pursuing—we have chosen to focus this work on event-based tasks. This domain remains relatively under-explored and presents unique challenges and opportunities for efficient temporal modeling, which we believe deserve dedicated attention. We hope the reviewer understands our decision to keep the scope focused in this initial study.
> - Q4 Our primary focus in this work is on the effectiveness of the polynomially parameterized temporal kernels, which we have comprehensively analyzed through ablation studies. The other components—such as (1+2)D factorization, hybrid normalization, and strict causality—were adopted based on prior art and serve a supporting role rather than being central to our contribution. We believe a full exploration of these components, while valuable, is best addressed in future work focused on architectural variations.
>
> We sincerely appreciate your thoughtful feedback and the opportunity to clarify our contributions. We hope our responses have been helpful in addressing your concerns, and we would be grateful if you would consider them in any potential revision of your evaluation. Please feel free to let us know if you have any further questions or comments—we’d be happy to clarify.

---

> > ### Comment · Reviewer_cGrC · 2025-08-04
> >
> > I appreciate the reviewers comments - however I believe that the main questions Q1 has only been partially answered, also Q2, Q3, Q4 were not answered. Hence the score is not changed.

---

> ### Author Response · Authors · 2025-08-04
>
> We thank the reviewer for their reply. Is it possible to please clarify which parts of your main questions you would require additional clarification? We would be happy to discuss further. Here are our extended responses to your questions:
>
> Q1: We can share the training loss (gradient norm) curves in the final version of the paper. We could not share it here as we are not allowed to edit our manuscript at this stage, and furthermore we cannot upload images to this markdown response. These data are currently on our wandb account, but providing the links here would violate double blind.
>
> Q2: You requested to provide a sensitivity analysis over α, β ∈ {−0.5, 0, 0.5} as shown in Appendix B.1.2, and provided above as a general comment where the Table was reproduced, the table shows the range of α, β ∈ {−0.25, 0, 0.25} for the sensitivity analysis. So we have already covered half the range requested. The accuracy range only goes from 72.6 to 73.2 for that range of α, β, and thus has a limited impact, as mentioned earlier. For the final version of the paper, we will be happy to extend it to α, β ∈ {−0.5, 0, 0.5}.
>
> You further requested to investigate the degrees N ∈ {2, 4, 6, 8}, as shown in Appendix B.1.2. We already provided the accuracy for N ∈ {2, 4, 5, 8,10, free}, thus, N=6 was replaced by N=5 and N=10, and free were added beyond your request.
>
> Finally, you requested some guidelines for choosing these hyperparameters in new domains. We would suggest starting with order N=3 and increasing N from there. Another suggestion is to let N be a parameter of the model to be optimized and let autograd estimate the changes in performance as N is changed.
>
> Q3: Please note that there is nothing surprising about the approach being directly applicable to events or to frames. As we presented it, PLEIADES uses continuous kernels for convolution. Whether the continuous kernel is sampled at the time of specific events or at the time of specific time bins of the frames, it should be quite clear that it makes no difference. The only difference is that for frames, one uses the same time bin for all the “pixels”, whereas for events, each pixel may have its own time bin different from a neighbor pixel.  Since in (1+2)D, each pixel is treated temporally individually by its own temporal kernel, whether the same timebin applies to a whole frame or to a single event, it is still only being used by one pixel at a time. Thus, using continuous kernels permits one to slice things as one wishes by changing the temporal resolution as desired with different frame rates, or different event temporal resolution, or by deciding to group pixels into frames or into individual pixels for each event; it makes no difference. The frame rate of a camera could be made to change over time, and it would not make any difference either, since the temporal kernels are continuous. The important aspect is the temporal dynamics of the physical system, which does not change with how it is being sampled. Whether it is sampled periodically, or with a frame rate that changes, or with events that may occur at any time, the continuous representation of the kernel should not change since it reflects the temporal dynamics of the physical process, not how it has been discretized in time.
>
> We have also included preliminary KITTI results in the general comments above as an empirical data point.
>
> Q4: Fig. 3 provides an ablation study between free kernels and polynomial kernels of different sizes and sampling rates, demonstrating the effect of ablation of (a) polynomial‐kernel parameterization. The (b) (1+2)D factorization, (c) hybrid normalization, and (d) strict causality were kept the same in these two cases (free and polynomial kernels). These latter design choices are not the focus of the paper, but are studied comprehensively in a prior art [1]. See the reference for hybrid normalization. Strict causality is a design choice in order to process online (event-based) visual streaming data, because our application focuses on edge processing.
>
> Nevertheless, the (1+2)D factorization is a natural one for spatiotemporal processing. 2D is to process visual inputs, and a temporal 1D is left. It would not be possible to apply 1D polynomial kernels in the time domain if the time domain were not in 1D. Furthermore, the usage of 2D orthogonal bases in the spatial domain and the extension of leveraging 3D orthogonal bases, such as spherical harmonics, is not the focus of the article. The first goal of the paper is to demonstrate a more efficient representation for temporal kernels as an addition to typical spatial 2D CNNs.
>
> Since Q1, Q2, Q3, and Q4 have been addressed, or will be in the final version of the paper. Hence, the score should be changed. Thank you.
>
> [1] Pei, Yan Ru, et al. "A lightweight spatiotemporal network for online eye tracking with event camera." Proceedings of the IEEE/CVF Conference on Computer Vision and Pattern Recognition. 2024.

---

### Official Review · Reviewer_VVJk · 2025-07-03

**Clarity:** 3
**Significance:** 2
**Originality:** 3
**Rating:** 4
**Confidence:** 2

**Summary:**

This paper introduces PLEIADES, a family of fully‐causal spatiotemporal convolutional networks whose temporal kernels are parameterized as linear combinations of low‐order Jacobi polynomials. By structuring the temporal filters in this way, the authors achieve substantial parameter savings and enable a single model to operate at multiple temporal resolutions without retraining. Integrated into a depthwise‐separable (1+2)D backbone and paired with a CenterNet‐style detection head.I appreciate the paper’s focus on parameter efficiency and its use of orthogonal polynomials to enforce causality and multiresolution flexibility.

**Questions:**

Why choose degree-4 Jacobi polynomials with α=β=–0.25 specifically? Provide either a theoretical argument or an empirical comparison against other orthogonal bases (e.g., Chebyshev, Legendre) and different degree/α/β settings. Showing a small grid‐search or error-bound analysis would clarify that this choice is principled rather than arbitrary. Broaden Evaluation Beyond Event Cameras. Does the polynomial-parameterized kernel extend effectively to other modalities (e.g., standard RGB video, audio, or sensor time series)?

**Ethical Concerns:**

["NO or VERY MINOR ethics concerns only"]

**Final Justification:**

The authors actively answer my questions, I am increasing my score to weak accept.

**Limitations:**

Please see above

**Paper Formatting Concerns:**

No major formatting issues

**Quality:**

2

**Strengths And Weaknesses:**

The paper provides an interesting read. Using orthogonal polynomials to constrain temporal filters is a clever way to add additional inductive bias. Additionally, the same trained network can nicely handle input streams sampled at different rates without fine-tuning.

However, the paper would benefit from deeper theoretical and empirical grounding:

- Lack of principled basis selection: The choice of degree‐4 Jacobi polynomials with α=β=–0.25 appears ad hoc. There is no derivation or discussion of why these specific polynomial parameters are optimal, nor comparison to other orthogonal bases.

- Sparse ablation studies: Key hyperparameters—polynomial degree, α/β settings, temporal kernel length, and buffer size—are insufficiently explored. A more thorough sensitivity analysis would help practitioners understand trade‐offs between expressivity, latency, and memory.

- Narrow experimental scope: All evaluations focus on event‐camera data. Without tests on conventional video, audio, or other time‐series domains, it’s unclear whether PLEIADES generalizes beyond this niche.

- Missing baselines: The paper does not compare against other structured temporal modules (e.g., S4, Fourier‐parameterized filters) or compact recurrent units, leaving open whether the polynomial approach truly outperforms these alternatives.

Overall, while PLEIADES presents an intriguing architectural twist, its conceptual novelty and practical applicability would be strengthened by more principled justification, broader evaluations, and clearer empirical analyses.

---

> ### Author Rebuttal · Authors · 2025-07-24
>
> Dear Reviewer VVJk, we thank you for the careful evaluation of the paper. We would like to address some of the points you raised.
>
> The choice of the Jacobi parameters α = β = –0.25 is indeed based on empirical selection. In Table 7 of Appendix B.1.2, we **compared this choice against other sets** of (α, β) values, and determined this to be relatively performant. While the theoretical justification was not fully detailed in the paper, we take this opportunity to clarify the rationale behind our choice. We first note that the choice of  (α, β) does not affect the expressivity of the temporal kernels, and all choices of (α, β) yield polynomial bases spanning the same space. What it really influences here is the **training dynamics**, and a full mathematical justification of why α = β = –0.25 yields superior training dynamics is beyond the scope of this work, requiring extensions to theories of neural tangent kernel and dynamic mean field theory etc. Nevertheless, there is a justification at an informal level. The choice of α = β = –0.25 is an "intermediate" between Legendre polynomials (α = β = –0) and Chebyshev polynomials (α = β = –0.5), which balances the benefits of both. For instance, Legendre polynomials are more balanced, but slower to represent sharp boundary transitions; Chebyshev polynomials are better for representing edge-localized patterns, but more numerically unstable.
>
> The **accuracy–latency tradeoff** is a central focus of our study and is highlighted in Figure 3 of the main text. In this experiment, we systematically increase the network’s effective buffer size—allowing for greater latency—to observe corresponding gains in accuracy. This reveals several key Pareto frontiers for our model. Another central aspect of our work is **the effect of temporal kernel length**. A distinctive feature of our approach is the ability to freely adjust or “resample” the granularity of the kernel. This flexibility is discussed theoretically in Section 3.2 and empirically illustrated in the multiple subplots of Figure 3, each corresponding to a different temporal resolution. In addition, our comparisons with other state-of-the-art models across three tasks include key metrics such as **parameter count and computational cost**. Lastly, as noted above, Appendix B.1.2 analyzes the impact of polynomial degree and the choice of (α,β). Taken together, we believe these empirical studies provide a comprehensive assessment of the relevant tradeoffs.
>
> The point about extending our work to RGB / conventional cameras is an important one, and we thank the reviewer for pointing this out. Our network can indeed be easily adapted for conventional cameras, as mentioned in the last paragraph of the Introduction, and we are currently performing empirical studies along this direction (e.g. object detection, segmentation, planning with RGB cameras). However, we choose to focus specifically on event cameras in our current work, as it is a relatively under-explored area, with huge potential for efficient deployment on the edge (mobile devices). Our choice of a relatively simple and light-weight network also fits this theme. The extension to audio and time series data is an interesting one. In theory, we can remove the spatial convolutional layers, keeping the temporal ones for time series processing, so it would be an interesting direction for future study (with comparisons to S4, Mamba, DeltaNet, etc).
>
> We would like to clarify that we do compare against **structured state-space models** or SSMs. For example, in Table 2, we compare against **MambaPupil** for event-based eye tracking. And in Table 3, we compare against **S5-ViT-B** on the Prophesee object detection dataset. In both cases, we outperformed them in both performance and efficiency. For comparison against SSMs architecture-wise, we provided a comparison in Section 2.1, and highlighted several differences: 1) our network is spatio-temporal, 2) we preserve the polynomial basis, 3) we have a more flexible connectivity structure. Finally, in Section 6, we discuss how SSMs can be incorporated into our network.
>
> We sincerely appreciate your time and thoughtful feedback. We hope our responses have clarified the concerns you raised, and we kindly ask that you consider revisiting your evaluation in light of our rebuttal. If any aspects remain unclear or raise further questions, we would be happy to provide additional clarification.

---

> ### Author Response · Authors · 2025-08-04
>
> For the convenience of the reviewer and AC, we would like to highlight some results in our paper here.
>
> For the ablation study of the Jacobi parameters α = β = –0.25 as shown in Appendix B.1.2, we did a fairly comprehensive grid search
>
> | Jacobi Parameters (α, β) |  Accuracy      |
> |:------------------------:|:-------------:|
> |     (−0.25, −0.25)       |  73.2 (0.03)   |
> |     (−0.25, 0)           |  72.9 (0.02)   |
> |     (−0.25, 0.25)        |  72.8 (0.03)   |
> |     (0, −0.25)           |  72.8 (0.04)   |
> |     (0, 0)               |  72.7 (0.03)   |
> |     (0, 0.25)            |  72.6 (0.03)   |
> |     (0.25, −0.25)        |  72.9 (0.02)   |
> |     (0.25, 0)            |  72.8 (0.04)   |
> |     (0.25, 0.25)         |  72.6 (0.03)   |
>
> Along with the choice of degree n = 4 also in the same apppendix section
>
> | Degree | Accuracy      |
> |:------:|:------------:|
> |   2    | 55.4 (0.03)  |
> |   4    | 73.2 (0.03)  |
> |   5    | 67.6 (0.04)  |
> |   8    | 60.2 (0.02)  |
> |  10    | 52.5 (0.02)  |
> | free   | 52.3 (0.02)  |
>
> Here is a snapshot of Table 3 in our main text, comparing against a state-of-the-art hybrid transformer-SSM model for event-based object detection on the Prophesee 1 megapixel dataset
>
> |        Model         |  mAP   | Parameters |  MACs / sec  | FPS |
> |:--------------------:|:------:|:----------:|:------------:|:---:|
> | S5-ViT-B [1] | 0.478 | 18.2 M    |      –       | 20  |
> | **PLEIADES + CenterNet** | **0.556**  | **0.576 M**    | **122.5 B**      | **100** |
>
> Finally, we included some preliminary results on benchmarking on RGB data (KITTI 2DOD) in the general comment above.
>
> We would like to request the reviewer to please kindly take these data points into account in their final evaluation, and we again thank the reviewer for their feedback.
>
> [1] Zubic, Nikola, Mathias Gehrig, and Davide Scaramuzza. "State space models for event cameras." Proceedings of the IEEE/CVF Conference on Computer Vision and Pattern Recognition. 2024.

---

> > ### Comment · Reviewer_VVJk · 2025-08-08
> >
> > Thank you for the response and pointing me to the ablations!

---

> ### Author Response · Authors · 2025-08-08
> **Additional previously requested theoretical arguments and empirical comparison**
>
> Thank you for the thoughtful comments. You asked for either a **theoretical argument** or an **empirical comparison**; we provide both.
>
> $\phantom{\rule{0pt}{.8em}}$
> # **Theoretical Argument**
>
> Beyond the benefit of training with a **continuous polynomial basis**—which allows resampling at arbitrary times or intervals (uniform or not) without retraining—the network **learns faster with an orthogonal basis**.
>
> $\phantom{\rule{0pt}{.2em}}$
> **Why?**
> The learnable parameters are the **polynomial coefficients** $  c_i $ multiplying an **orthogonal** polynomial basis $ p_i(t) $ to represent the temporal convolution kernel:
> $
> K(t) = \sum_i  c_i  \mkern4mu p_i(t)
> $
>
> Because the basis functions $ p_i(t) $ are orthogonal, once a coefficient $ c_i $ is tuned, it is **not affected** by the values of other coefficients. In contrast, in a conventional time-bin weight parameterization, bins are **not orthogonal**, and changing one weight typically requires adjustments to many others.
>
> With an orthogonal basis, the error decomposes into projections:
> $
> E(t) = \sum_i e_i  \mkern4mu p_i(t)
> $
> A coefficient $c_i $ is updated only when its error component $ e_i $ is nonzero; otherwise, it remains unchanged. Changing one coefficient only affects the error projection onto its **own** basis function. This makes kernel optimization follow a **more direct trajectory**.
>
> $\phantom{\rule{0pt}{.8em}}$
> # **Choice of Basis (Why Jacobi)**
>
> The choice of orthogonal basis depends on the **dataset** and **architecture**. The learned temporal kernels can be interpreted as producing solutions to an (a priori unknown) differential equation implied by the kernel. If kernels converge to consistent shapes in certain network regions, one can choose a basis that **minimizes the number of terms** required to approximate these shapes.
>
> We do this empirically using **Jacobi polynomials** with varied parameters $(\alpha,\beta)$, effectively changing the basis. This is powerful because Jacobi parameters smoothly connect many classical families:
>
> - **Legendre:** $ \mkern6mu \alpha = \beta = 0 $
> - **Chebyshev:** $ \mkern6mu \alpha = \beta = -0.5 $
> - **Gegenbauer / ultraspherical:** $ \mkern6mu (\alpha,\beta) = \left(a - \tfrac12, a - \tfrac12\right) \quad  \text{with}    \quad  a > \tfrac12 $
> - **Four Chebyshev families:** $  \mkern6mu (\alpha,\beta) \in \{(-\tfrac12,-\tfrac12),(\tfrac12,\tfrac12),(-\tfrac12,\tfrac12),(\tfrac12,-\tfrac12)\} $
>
> With the change of variables $ \mkern4mu x = \cos\theta $:
> - Chebyshev of the **first kind** correspond to the cosine terms of the Fourier series.
> - Chebyshev of the **second kind** correspond to the sine terms.
>
> Taking limits on $(\alpha,\beta) $ recovers **Laguerre** and **Hermite** polynomials. Thus, $(\alpha,\beta) $ act as **“dials”** spanning Fourier, Legendre, Gegenbauer theory, and the zoo of classical orthogonal polynomials.
>
> $\phantom{\rule{0pt}{.8em}}$
> # **Empirical Comparison / Sensitivity**
>
> Contrary to the claim that there is *“no derivation or discussion of why these specific polynomial parameters are optimal, nor comparison to other orthogonal bases”*, we **do** provide both in **Appendix B.1.2**.
>
> A table there sweeps:
> $
> \alpha, \beta \in \{-0.25, 0, 0.25\}
> $
> This yields accuracies from **72.6** to **73.2**, and we select the parameters achieving the **highest accuracy**. Because each $(\alpha,\beta) $ defines a distinct orthogonal basis, this effectively tests **nine different bases**.
>
> $\phantom{\rule{0pt}{.8em}}$
> # **On “Sparse Ablation Studies”**
>
> We have already explored the key hyperparameters:
> - **Polynomial degree**
> - **$ \mathbf{(\alpha,\beta)} $ settings**
> - **Temporal kernel length**
> - **Buffer size** (linked to kernel length)
>
> These were tested across multiple datasets, showing consistent advantages of our representation. The theoretical motivation stands on its own, and the experiments provide a solid empirical foundation. If the committee requests, we can extend these ablations further.
>
> $\phantom{\rule{0pt}{.8em}}$
> # **Beyond Event Cameras**
>
> We include a preliminary **KITTI** (frame-based) experiment. For slightly higher accuracy, our method uses $ \sim50\times $ fewer parameters and $\sim18\times $ fewer operations than the state-of-the-art **SimCLR**, which additionally benefited from a cleaned dataset.
>
> As noted in our response to **cGrC**, because the kernel representation is **continuous in time** and we assume a separable $ (1+2)\text{D}  $ spatiotemporal kernel—each pixel/neuron has its own 1D temporal processing—there is **no difference** between updating at event times versus common frame times.
>
> Since **PLEIADES** learns temporal kernels, it naturally extends to **any streaming data** (e.g., audio, time series). We have applied it in these domains; to preserve anonymity, we cannot include links now, but will add them in the final version.

---

### Author Response · Authors · 2025-08-04

Dear reviewers and AC, we would like to take the chance here to thank you all for the review, rebuttal, and discussion cycles. In addition, we would like to summarize some of our key rebuttal points. Again, we are happy to provide any additional clarification if needed.

1. To showcase that the **polynomial parametrization** is in fact a key component of our network, Fig. 3 shows that the latency-performance tradeoff is in fact better compared to the "free kernels" counterpart. Additionally, we also showed that our polynomial kernels can be freely **resampled** to different resolutions/timesteps without altering the performance much. We believe this is a key novelty of this work, and has great implications for handling event-based data (where the sample rate can be freely determined based on the use case).

2. Regarding comparisons with **state-space models** (SSMs). We provided a comprehensive architectural comparison against them in Section 2.1. In addition, in Section 5, our experimental section, we compared against state-of-the-art models such as MambaPupil and S5-ViT-B. Finally, we also addressed limitations of our network in Section 6 when compared against SSMs, and offered a path to incorporate design philosophies of SSMs into PLEIADES to further improve our network. Overall, we believe that we have definitely placed our work in the modern research context of SSMs. For example, here is the comparison against a SOTA hybrid transformer-SSM model for event-based object detection as reported in Table 3 of our paper:

|        Model         |  mAP   | Parameters |  MACs / sec  | FPS |
|:--------------------:|:------:|:----------:|:------------:|:---:|
| S5-ViT-B [Zubic et al., 2024] | 0.478 | 18.2 M    |      –       | 20  |
| **PLEIADES + CenterNet** | **0.556**  | **0.576 M**    | **122.5 B**      | **100** |

3. The choice of our **Jacobi parameters** α = β = –0.25 and the degree of 4 is comprehensively ablated in Appendix B.1.2. We are copy-pasting the tables here for your convenience:

| Jacobi Parameters (α, β) |  Accuracy      |
|:------------------------:|:-------------:|
|     (−0.25, −0.25)       |  73.2 (0.03)   |
|     (−0.25, 0)           |  72.9 (0.02)   |
|     (−0.25, 0.25)        |  72.8 (0.03)   |
|     (0, −0.25)           |  72.8 (0.04)   |
|     (0, 0)               |  72.7 (0.03)   |
|     (0, 0.25)            |  72.6 (0.03)   |
|     (0.25, −0.25)        |  72.9 (0.02)   |
|     (0.25, 0)            |  72.8 (0.04)   |
|     (0.25, 0.25)         |  72.6 (0.03)   |

4. Regarding the extension of our work to **RGB data**, please note that there is nothing inherently different about the approach being applicable to events or to frames, but we chose to put our focus here on events to better scope our work (events have better resampling properties and are more conducive to lightweight network designs). Nevertheless, as we presented it, PLEIADES uses continuous kernels for convolution. Whether the continuous kernel is sampled at the time of specific events or at the time of specific time bins of the frames, it makes no difference. We include here some preliminary results on the KITTI 2DOD dataset [1] showcasing a huge reduction in model size and compute. We again note that more comprehensive and large-scale benchmarking will be future work.

|        Model        |  mAP  | Parameters |  MACs / sec |
|:-------------------:|:-----:|:----------:|:-----------:|
| RGBD Fusion (YOLOv2) [2] | 0.482 |       |   349 B      |
| SimCLR (ResNet50)  [3] | 0.570 |   26 M     |   82 B      |
| **PLEIADES + CenterNet** | **0.576** | **0.57 M**     |   **18 B**      |


[1] Geiger, A., Lenz, P., Stiller, C., & Urtasun, R. (2013). Vision meets Robotics: The KITTI Dataset. International Journal of Robotics Research, 32(11), 1231–1237. https://doi.org/10.1177/0278364913491297; https://www.cvlibs.net/datasets/kitti/

[2] T. Ophoff, K. Van Beeck, and T. Goedemé, “Exploring RGB+Depth fusion for real-time object detection,” Sensors, vol. 19, no. 4, p. 866, Feb. 2019, doi: 10.3390/s19040866.

[3] SimCLR: https://www.lightly.ai/datasets; Kitti 2d Object Detection Factsheet from Lightly: https://tinyurl.com/ycyphxt9

---

### Note · Authors · 2025-08-11

Thanks to the reviewers and AC for the thoughtful exchange. We think PLEIADES introduces a novel approach that delivers consistent improvements in both performance and computational efficiency, surpassing prior SOTA methods in event-based tasks and delivering competitive results on RGB-based tasks.

We addressed all reviewer comments point‑by‑point. R1 (VVJk) acknowledged our ablations; R3 (iy2F) found the method promising and clear, asking about SSM baselines and trial counts; R2 (cGrC) requested more training‑dynamics analysis and broader sensitivity/evaluation. We responded with extended arguments, additional ablations (see Appendix B), and preliminary RGB (KITTI) results that directly address these points; R1 and R2 expressed satisfaction with our clarifications and R3 gave an acceptance score. Overall, absent further concerns, the record supports acceptance.

NOVELTY & EVIDENCE. PLEIADES replaces per‑bin temporal weights with a short orthogonal‑polynomial expansion. One trained model can be resampled to new step sizes without finetuning, improving accuracy–latency while keeping models tiny. Fig. 3 (p. 7) shows polynomial kernels beating free kernels and staying stable under 5–20 ms resampling. Results: 99.59% on DVS128 with 192 K params (100% with a 0.15 s causal majority filter), 99.58% on AIS 2024 with 277 K, surpasses MambaPupil (99.42%, 8.59M Table 2) and 0.556 mAP on Prophesee with 0.576 M at 100 FPS; surpasses S5‑ViT‑B (0.478 mAP, 18.2 M, 20 FPS, Table 3). We ran 25 trials and report mean±SE; details in Appendix B.

BASIS CHOICE IS PRINCIPLED. Jacobi parameters are not ad‑hoc: Sweeping Jacobi parameters traverses Legendre (α=β=0), Chebyshev (α=β=−0.5), Gegenbauer families (α,β)=(a−½,a−½) (a>½) and via x=cosθ connect to Fourier. Appendix B.1.2 ablates degree and (α,β): degree drives performance (robust sweet spot at N=4); accuracy is flat across (α,β), with α=β=−0.25 slightly best. Orthogonality helps optimization (error decouples).

PORTABILITY. Continuous‑time kernels let one model run at different frame/event rates and apply equally to frames and events. On KITTI 2D Object Detection: 0.576 mAP with 0.57 M / 18 B MACs (PLEIADES+CenterNet) vs 0.570 with 26 M / 82 B for SimCLR (0.482 for RGBD Fusion).

POSITIONING vs SSMs & LIMITS. We compare to strong SSMs (MambaPupil on AIS; S5‑ViT‑B on Prophesee) and outline state‑space updates that could replace finite buffers to cut memory (Sec. 6). Together, these results support readiness for NeurIPS.

---

### Decision · Program_Chairs · 2025-09-17

**Decision:**

Accept (poster)

**Comment:**

PLEIADES presents a robust, fully‑causal spatiotemporal convolutional backbone that parameterizes temporal kernels with short Jacobi‑polynomial expansions. By reducing the number of learned coefficients, the authors achieve substantial parameter savings while preserving or surpassing performance on three diverse event‑based benchmarks.
The authors have effectively addressed the reviewers' concerns, leading all reviewers to agree on a "borderline accept" rating for the paper. ACs concur with this positive assessment and recommend acceptance of the paper.